# Determination of Chemical Stability of Two Oral Antidiabetics, Metformin and Repaglinide in the Solid State and Solutions Using LC-UV, LC-MS, and FT-IR Methods

**DOI:** 10.3390/molecules24244430

**Published:** 2019-12-04

**Authors:** Anna Gumieniczek, Anna Berecka-Rycerz, Tomasz Mroczek, Krzysztof Wojtanowski

**Affiliations:** 1Department of Medicinal Chemistry, Medical University of Lublin, Jaczewskiego 4, 20-090 Lublin, Poland; anna.berecka@umlub.pl; 2Department of Pharmacognosy with Medicinal Plant Unit, Medical University of Lublin, Chodźki 1, 20-093 Lublin, Poland; tmroczek@pharmacognosy.org (T.M.); krzysztofkamilw@gmail.com (K.W.)

**Keywords:** metformin, repaglinide, kinetics of degradation, degradation products, excipients, LC-UV, LC-MS and FT-IR

## Abstract

Firstly, metformin and repaglinide were degraded under high temperature/humidity, UV/VIS light, in different pH and oxidative conditions. Secondly, a new validated LC-UV method was examined, as to whether it validly determined these drugs in the presence of their degradation products and whether it is suitable for estimating degradation kinetics. Finally, the respective LC-MS method was used to identify the degradation products. In addition, using FT-IR method, the stability of metformin and repaglinide was scrutinized in the presence of polyvinylpyrrolidone (PVP), mannitol, magnesium stearate, and lactose. Significant degradation of metformin, following the first order kinetics, was observed in alkaline medium. In the case of repaglinide, the most significant and quickest degradation, following the first order kinetics, was observed in acidic and oxidative media (0.1 M HCl and 3% H_2_O_2_). Two new degradation products of metformin and nine new degradation products of repaglinide were detected and identified when the stressed samples were examined by our LC-MS method. What is more, the presence of PVP, mannitol, and magnesium stearate proved to affect the stability of metformin, while repaglinide stability was affected in the presence of PVP and magnesium stearate.

## 1. Introduction

Metformin hydrochloride, 3-(diaminomethylidene)-1,1-dimethylguanidine (Figure 1A), is an oral antidiabetic drug that is used for the treatment of type 2 diabetes, particularly for overweight or obese patients with normal kidney function. It reduces the serum glucose level by several actions, notably through non-pancreatic mechanisms without increasing insulin secretion. This agent mainly works by reducing gluconeogenesis and opposing glucagon-mediated signaling in the liver and, to a lesser extent, by increasing glucose uptake in skeletal muscles [1]. Repaglinide, (*S*)-(+)-2–ethoxy-4-[2-(3-methyl-1-[2-(piperidin-1-yl)phenyl]butylamine)-2-oxoethyl] benzoic acid (Figure 1B), is a non-sulfonylurea insulin secretagogue that produces a hypoglycemic effect by stimulating insulin secretion from the pancreatic β-cells, but, in contrast to the sulfonylureas, it acts via different binding sites on the β-cells [2]. Metformin and repaglinide are now available in fixed-dose combinations because of their complementary actions in diabetes.

The stability of the drug substance or product is defined as capacity to remain within established specifications, i.e., to maintain identity, purity, and strength until the expiry date [3]. Stability testing provides information regarding how the quality of the drug substance or product varies with time under temperature, humidity, oxygen, and light. Thus, knowledge from stability studies can be used during manufacturing process, selecting proper packaging, storage conditions, and product shelf-life [4,5]. In turn, the stress experiments allow for the generation of related compounds in much shorter time. In addition, the stressed samples can be used to develop the stability-indicating quantitative methods. Subsequently, they can be applied in the analysis of samples from accelerated and long term stability studies [6].

Several stability-indicating HPLC and UHPLC methods were reported for the determination of metformin, mainly in combination with other antidiabetic drugs, like sitagliptin, linagliptin, alogliptin, vildagliptin, dapagliflozin, empagliflozin, pioglitazone, and glimepiride [7,8,9,10,11,12,13,14,15,16,17,18]. Additionally, stability-indicating HPTLC methods have been reported for similar purposes [19,20]. The selectivity of the above methods was confirmed by analyzing the samples of metformin that were subjected to acidic (0.1–2 M HCl), alkaline (0.1–2 M NaOH), and oxidative (3–30% H_2_O_2_) conditions, as well as to UV light or natural sunlight and high temperature (105 °C). 

Several stability-indicating HPTLC [21,22,23,24] and UHPLC [25,26,27] methods were described for the determination of repaglinide. Their selectivity was proved by analyzing the samples of repaglinide stressed with 0.1–1 M NaOH, 0.1 M HCl, 6–10% H_2_O_2_, and direct sunlight. Two reports on the simultaneous determination of metformin and repaglinide were also reported [28,29].

The related substances of metformin are described in European Pharmacopoeia [30] as impurities A–F (Imps A–F) (Table 1). Therein, ion-exchange and reversed phase chromatography with pre-column derivatization are recommended for the determination of Imp A (*N*-cyanoguanidine) and Imp F (*N*-methylmethanamine), respectively. 

Five related compounds (Imps A–E) were described (Table 2) in the official monograph of repaglinide [30]. The separation of Imps A–D can be achieved while using gradient chromatography on alkyl-bonded silica gel and mobile phases with high water content. In addition, chiral gradient chromatography is recommended for the determination of Imp E (isomer of repaglinide). 

Bearing previously published reports on the chemical stability of metformin and repaglinide in mind, we performed further experiments. The first goal was to compare their stability in different environments, examining percentage degradation as well as the kinetics of degradation. For these purposes, degradation under extraordinary pH (0.01–0.1 M HCl and 0.01–0.1 M NaOH), oxidative conditions (3–30% H_2_O_2_) at 70 °C as a function of time was investigated, using the new validated stability-indicating LC-UV method. The next goal was to detect the degradation products of metformin and repaglinide and elucidate the possible degradation patterns, by analyzing the stressed samples while using our LC-MS method.

It is known that some excipients present in the final drug products may contribute to the instability of active pharmaceutical substances due to their acidic, basic, or reducing properties, as well as due to the presence of their own impurities [32]. Meanwhile, the chemical stability of metformin and repaglinide in the presence of excipients has not been extensively studied. Therefore, an additional purpose of the present study was to examine their interactions with polyvinylpyrrolidone (PVP), mannitol, magnesium stearate, and lactose, at high temperature/humidity (70 °C/70% RH), and under UV/VIS light (300–800 nm). The FT-IR spectra of the solid stressed samples were examined in order to detect these interactions.

## 2. Results and Discussion

### 2.1. Elaboration of Quantitative LC-UV Method

The chromatographic conditions were optimized to achieve the best peak shapes and the best resolution for metformin, repaglinide, and their degradation products. Different mobile phases containing acetonitrile, methanol, water, and ammonium acetate were examined. It was found that methanol was not effective for sufficient separation of the peaks. In addition, it was still difficult to reduce the peak tailing, even when different volumes of methanol were added to the mobile phase. Thus, acetonitrile as an organic solvent was tried in different combinations with water and acetate ammonium solution, which led to better separation and peak shapes. Finally, the mobile phase containing 0.01 M acetate ammonium and acetonitrile (60:40, *v/v*) was selected for obtaining well defined and resolved peaks with mean retention times of 2.55 and 7.68 min., for metformin and repaglinide, respectively (Figure 2). Some asymmetry of the peaks still occurred, due to the high amount of differences in the polarity of the compounds, but the obtained asymmetry factors were still within the acceptable limits of 0.8 and 1.5 (Table 3).

### 2.2. Validation of Quantitative LC-UV Method

The specificity of the method was confirmed as the ability of the method to determine the non-degraded metformin and repaglinide in the presence of their degradation products. Under the above chromatographic conditions, the peaks of degradation products were rather low and appeared at the beginning of the solvent front. In addition, they were sufficiently separated from the peaks of interest (Figure 2). Specificity was also confirmed as the ability of the method to determine metformin and repaglinide in the presence of the excipients present in respective powdered tablets (data not shown).

The robustness of the method was estimated under slightly changed analytical parameters, i.e., with the flow rate of the mobile phase of 1.0 ± 0.2 mL/min., acetonitrile content in the mobile phase of 40 ± 5%, and the detection wavelength of 235 ± 3 nm. The uniformity of the obtained peak areas, retention times (t_R_), and resolution between the peaks of interest (R_S_) confirmed the robustness of the method (Table 4).

The peak areas of metformin and repaglinide were plotted against the corresponding concentrations of the drugs, to construct the calibration equations. The linearity of the method was confirmed in the range 0.015–0.09 mg/mL for both drugs, giving the regression equations y = 44.4609x + 0.1877 for metformin and y = 9.2725x + 0.0039 for repaglinide, where y was the peak area and x was the concentration of the drug in mg/mL. The mean (±SD) determination coefficient (R^2^) was 0.9993 (±0.0003) for metformin and 0.9995 (±0.0003) for repaglinide. The LOD and LOQ, as calculated from the standard deviation of the intercept and slope of the regression lines, were 0.001 and 0.004 mg/mL for metformin, and 0.0006 and 0.002 mg/mL for repaglinide. The RSD values for intra-day and for inter-day precision were in the ranges 1.07–1.60% for metformin and 0.67–1.15% for repaglinide, which confirmed that the applied method was sufficiently precise. When metformin was assayed in tablets, the recovery ranged from 99.81 to 100.98%, while it ranged from 98.48 to 101.63% for repaglinide (Table 3).

### 2.3. Percentage Degradation of Metformin and Repaglinide in Solutions

In acidic medium, the degradation of metformin reached 5.73 and 6.73% after 240 min., in 0.01 M and 0.1 M HCl, respectively. At the same time, metformin was slightly sensitive to oxidative conditions, because its degradation in 0.3 and 3% H_2_O_2_ was determined as 6.58 and 7.95%. However, metformin turned out labile in the alkaline medium, where its degradation reached 9.11% (0.01 M NaOH) and 60.92% (0.1 M NaOH) (Table 5). 

According to some reports from the literature, metformin occurred similarly labile in acidic and alkaline media with the degradation of 5.97–7.73% and 4.66–6.47%, respectively [8,12]. On the other hand, the different results were also reported, according to which the degradation of the drug was significantly higher in alkaline (36.70–42.72%) than in acidic (0.5–15.24%) conditions [9,13]. Similarly, our study showed much higher degradation in 0.1 M NaOH than in 0.1 M HCl, i.e., 60.92 versus 5.63%. Thus, the high sensitivity of metformin to alkaline conditions seemed to be proved. Some small differences were noted when the sensitivity of metformin to oxidative conditions was studied. Two previously published papers showed that metformin degraded at the level of 3.64–20.34%, depending on the temperature used [8,12]. In the present study, the oxidative degradation in the range of 6.58–7.95% was stated, which indicated slight sensitivity of the drug towards oxidants (Table 5).

According to the literature, significant degradation of repaglinide was observed when the drug was subjected to acidic (35.8%) and alkaline (41.9%) conditions. At the same time, repaglinide was stabile in the oxidative medium, where the degradation was below 0.08% [25]. Patel et al. presented quite different results [26] for acidic and alkaline conditions, where the degradation was negligible. On the other hand, the study from Sharma and Sharma [27] showed the high degradation of repaglinide by alkaline hydrolysis (85.1%), high degradation by oxidation (40.7%), and significant degradation by acidic hydrolysis (28.7%). Our results proved that repaglinide was significantly sensitive to acidic and oxidative conditions. When 0.01 M and 0.1 M HCl were used, the degradations of repaglinide were 19.93 and 38.32%. As far as oxidative conditions were concerned, the degradation percentages of repaglinide were 9.16 and 21.75 in 0.3 and 3% H_2_O_2_, respectively. However, the drug occurred more stabile in the both 0.01 M and 0.1 M NaOH, where its degradation was in the range 6.13–7.24% (Table 5). What is more, in the study of Joshi et al. [28], where the stability of metformin and repaglinide were examined in the same conditions, metformin was shown to be labile in alkaline medium, while repaglinide was sensitive to acidic conditions, similarly to our results. As far as oxidative degradation was concerned, both drugs were reported to be rather resistant. Unfortunately, the authors did not report the percentage levels of degradation for individual drugs.

### 2.4. Kinetics of Degradation

Our study showed stronger correlations (higher R^2^ values) for the plots of logarithms of concentration of non-degraded metformin versus time of degradation, which confirmed the first order kinetics for its degradation. The calculated t_0.5_ values varied from 50.13 h (0.3% H_2_O_2_), 16.74 h (0.01 M HCl, 0.1 M HCl, and 3% H_2_O_2_) through 12.50 h (0.01 M NaOH) to 0.97 h (0.1 M NaOH), which confirmed the lowest stability of metformin in a strong alkaline medium (Table 5). In the study of Mohamed et al. [10], some kinetic parameters for the degradation of metformin were estimated while using HPTLC method. It was found that acidic degradation (0.2 M HCl at 75 °C) was of a zero-order reaction with t_0.5_ 8.85 h. The zero-order reaction rate was also calculated for degradation in 30% H_2_O_2_ with t_0.5_ equaling 60.40 h. As far as alkaline degradation was concerned (0.2 M NaOH at 75 °C), it obeyed the pseudo-first-order reaction rate with t_0.5_ equal 2.91 h, which confirmed our results. Regarding the degradation of repaglinide, it also followed the first order kinetics. The calculated t_0.5_ values were 8.37–3.58 h (0.3 and 3% H_2_O_2_), 16.74–12.50 h (0.01 and 0.1 M NaOH), and 4.18–2.09 h (0.01 and 0.1 M HCl), which confirmed the lowest stability of repaglinide in acidic and oxidative conditions (Table 5). 

Weakly basic metformin (pKa 2.8 and 11.5) is ionized in pH range 4–10 when acidic repaglinide (pKa 4.19 and 5.78) is ionized at higher pH values. Taking our results presented above into consideration, it could be supposed that the non-ionized forms of metformin and repaglinide are more susceptible to degradation. As was described above, there was only one paper in the literature regarding the kinetics of degradation in respect to metformin, while we did not find any similar report concerning repaglinide. Thus, the results that are presented here supplemented the literary resources in this area.

### 2.5. Identification of Degradation Products

In two previous studies concerning metformin, the protonated precursor ion [M + H]^+^ at *m/z* 130.1 was observed, while the most stable ion product was detected at *m/z* 60.1, due to the loss of guanidine moiety. Thus, the above transition was used for the quantification of the drug in biological materials [33,34]. The biodegradation of metformin was examined in the study of Markiewicz et al. [31]. It was observed that the signal of parent drug at *m/z* 130 gradually decreased and the *m/z* 103 matching the guanylurea that appeared (DP1) (Table 1). Thus, the decomposition of metformin by removing of two methyl groups at the terminal nitrogen atom, leading to the formation of biguanide, and further guanylurea was suggested. Another pathway could be by removing of two urea molecules, giving dimethylguanidine, and finally dimethylamine. Figure 3 shows these two pathways for degradation of metformin. 

Different processes induced the degradation of metformin, i.e., direct photolysis (UVC), photocatalysis (TiO_2_/UVC), ozonation, and chlorination, in the study of Quintao et al. [35]. Subsequently, all of the samples were analyzed by LC/HRMS, by which five products were detected, and the molecular formulae for all of them were proposed. However, it was not possible to identify any of these degradants. Besides, the degradation of metformin in alkaline medium (1 M NaOH) was reported, whereby two degradants, i.e., 1,3,5-triazine-2,4,6-triamine (DP2=Imp D) and 1-methylbiguanide (DP3=Imp E) were identified (Table 1), while using the HPLC method [17]. In the present study, the most interesting results were obtained for the samples of metformin that were degraded in oxidative conditions, although the percentage level of degradation was not the highest. Firstly, the formation of 1-methylbiguanide (DPIV=Imp E=DP3) and biguanide (DPII) was documented. Thus, removing of one methyl group or both methyl groups at the terminal nitrogen atom was proved the decomposition of metformin. Besides, two new products of degradation, i.e., 1,2,4-triazole derivatives (DPI and DPIII), were proposed as the results of cyclization processes that had not been reported so far. Figure 4 shows the proposed degradation pathways, while Figure 5 shows the proposed degradation pathways.

In the previous study regarding repaglinide, the protonated precursor ion [M + H]^+^ at *m/z* 453.2 was observed, while the most stable ion product was detected at *m/z* 230.4. Monitoring the above transition was used for the quantification of repaglinide in rat plasma [36]. In the study of Kancherla et al. [37], seven impurities of repaglinide (degradation products (DPs) 1–7) were isolated from the crude sample of repaglinide and their chemical structures were characterized based on LC-ESI/MS (Table 6). It is worth mentioning that none of them was documented in the pharmacopoeial monograph of repaglinide.

In addition, one report from the literature [38] described the stress degradation of repaglinide and reported six degradation products (DPs 8–13). Four DPs (8–11) were formed under acidic conditions (1M HCl), two DPs (8,9) under basic conditions (1M NaOH), while three DPs (11–13) were formed under oxidative conditions (30% H_2_O_2_). The DP10 product was identified as Imp C, according to European Pharmacopoeia [30].

In our study, fifteen degradation products of repaglinide (DPs I–XV) were detected and identified in the stressed samples. Thus, the decomposition of repaglinide through the amide bond cleavage leading to the formation of DPIII (Imp C) was confirmed in acidic, alkaline, and oxidative conditions. Further, isomerization leading to form DPV (Imp E) could be proposed for acidic and alkaline conditions. In addition, previously observed decarboxylation, followed by hydroxylation, was confirmed for acidic medium, and the structural formulae similar to DP9 were proposed as DPIV. What is more, nine new compounds (DPI and DPs VIII–XV) were detected and identified after degradation in acidic and oxidative conditions. As a consequence, new ways of decomposition of repaglinide could be proposed as a result of cyclization (DPI) or cyclization preceded by the amide bond cleavage (DPs VIII–X). Subsequently, new degradants (DPs XI–XV) were identified in the samples of repaglinide that were stressed with H_2_O_2_. It is worth mentioning that such degradation pathways have not been reported so far. Figure 6 shows the chromatograms of repaglinide that were treated with 0.1 M HCl, 0.1 M NaOH, and 3% H_2_O_2_, while Figure 7, Figure 8 and Figure 9 show the proposed degradation pathways.

### 2.6. Degradation in a Solid State and Impact of Excipients

The FT-IR spectrum of metformin showed significant bands at 3365 and 3289 cm^−1^ due to N-H stretching and at 3147 cm^−1^ due to C-H stretching vibrations in the present study. In addition, characteristic bands at 1618 and 1549 cm^−1^ due to C=NH stretching, at 1445 cm^−1^ due to C-H bending and at 1060 cm^−1^ due to C-N stretching were clearly seen (Figure 10A). 

The spectrum of metformin that was stressed with high temperature and humidity did not show any significant changes when compared with that of the untreated one. However, some interesting results were obtained for binary mixtures of metformin with some excipients, as well for the untreated for the stressed samples. When the non-stressed mixture of metformin with PVP was examined, the spectrum showed the overlapping of the bands due to the C=O group of PVP and the C=NH group of metformin. The deteriorating of the band of PVP to a lower wavenumber from 1654 to 1636 cm^−1^ was also observed. Therefore, it was supposed that hydrogen bond formation between the amine group of metformin and carbonyl group of PVP occurred. These changes were accelerated after stressing the mixture with high temperature and humidity (Figure 10B). The literature also suggested the formation of hydrogen bonds between metformin and PVP [39]. On the other hand, other authors reported no interactions between metformin and PVP [40,41]. When metformin was mixed with mannitol, a decrease of the band of metformin at 3289 cm^−1^ due to the N-H stretching vibrations observed in the spectrum. This change was also observed after stressing the mixture with high temperature and humidity (Figure 10C). Therefore, we supposed that the amine group of the drug was affected. When the spectrum of the untreated mixture of metformin with magnesium stearate was examined, all of the main bands of the drug and excipient, with intensities proportional to their quantity in the mixture, were seen. However, the band of metformin at 3147 cm^−1^ due to C-H stretching vibrations decreased visibly after stressing the mixture with high temperature and humidity (Figure 10D). Thus, the observed interactions between metformin and mannitol, magnesium stearate, or PVP can affect the chemical stability of the drug and accelerate its degradation in the final formulations. It is known from the literature that magnesium stearate that are present in formulations can change the pH of their micro-environments to higher values (because of magnesium oxide impurity) and, consequently, accelerate the hydrolysis of some labile drugs. While taking the high sensitivity of metformin to alkaline degradation into consideration, it is not out of the question. What is more, some interactions between magnesium stearate and the compounds containing the amine group were reported earlier [42]. In addition, the sensitivity of metformin to oxidative conditions should be considered while bearing the potential presence of highly reactive peroxides as synthetic by-products of PVP in mind [32]. On the other hand, the lack of interactions between metformin and magnesium stearate was reported earlier [40,41], which confirmed the necessity of further studies in this area. As far as the impact of lactose was concerned, we did not observe any significant changes in the respective mixture, likewise for the untreated and the stressed samples, which was in agreement with some previously published data [40]. On the other hand, Santos et al. [41] observed alterations in the temperature of the melting peak of metformin mixed with lactose. Thus, potential interactions and the Maillard´s reaction were suggested with the participation of the amine group of the drug. 

The spectrum of individual metformin as well as the spectra of its all binary mixtures did not show any significant changes after the exposition as far as the impact of UV/VIS light in the solid state was concerned.

In the present study, characteristic absorption peaks of repaglinide were seen at 3305 cm^−1^ due to N-H vibrations and at 1683 cm^−1^ due to C=O stretching vibrations (the carboxylic group). Additionally, C-H stretching vibrations were seen at 2933 cm^−1^, C-O stretching vibrations of aryl ethers at 1210 cm^−1^, and C-N stretching vibration of the tertiary aromatic amine at 1383 cm^−1^. In the amide group, a sharp bond at 1628 cm^−1^ due to N-H bending vibrations and a combination of bonds of N-H deformation and C-N stretching vibrations were recorded at 1566 cm^−1^ (Figure 11A). 

The spectrum of individual repaglinide treated with high temperature and humidity did not show any significant changes when compared with that of the non-stressed drug. However, we observed some interesting interactions in the presence of some excipients, for both the untreated and stressed samples. First of all, disappearing the band of repaglinide at 1628 cm^−1^ due to N-H bending vibrations in the amide group was found in the case of PVP, even without any stressing (Figure 11B). What is more, additive changes were observed when that mixture was treated with high temperature and humidity, i.e., bands of repaglinide at 1683 and 1628 cm^−1^ and the band of PVP at 1654 cm^−1^ overlapped to one broad band at 1634 cm^−1^ (Figure 11C). However, the observed results were in the contrary with those that were reported in the literature, where no interactions between repaglinide and PVP were found when the respective spectrum was examined [43]. In the presence of magnesium stearate, we observed the disappearance of the band of repaglinide due to C-H stretching vibrations at 2933 cm−1, especially after high temperature/humidity stress. As a result, only one sharp band at 2915 cm^−1^ due to C-H stretching vibrations of magnesium stearate was identified (Figure 11D). The observed interactions between repaglinide and PVP, and between repaglinide and magnesium stearate, could be important as far as the chemical stability of repaglinide in the final drug products is concerned. Firstly, the drug was shown to be prone to oxidation. Thus, the presence of peroxide by-products in PVP should be taken into consideration [32]. Secondly, the interactions between the drugs containing the carboxylic group and magnesium stearate were reported in the literature and they could not be negligible [42]. However, to some extent, these results were different than those that were reported in the literature, where no interactions between repaglinide and magnesium stearate were stated [44]. Thus, it is supposed that different conditions for the stability studies were used.

The rest of excipients that were used in the present study did not show any interactions, because all of the characteristic bands of repaglinide were found in the spectra of respective binary mixtures, with intensities that were proportional to the contents of ingredients. This indicated the absence of chemical interactions between repaglinide and mannitol or lactose. Finally, the spectra of individual repaglinide as well as the spectra of all its binary mixtures, did not show any significant changes after irradiation with UV/VIS light.

## 3. Materials and Methods

### 3.1. Materials and Standards

The used chemicals were as follows: metformin from Polpharma S.A. (Starogard Gdanski, Poland) and repaglinide from Sigma-Aldrich (St. Louis, MI, USA), ammonium acetate, lactose, magnesium stearate, mannitol, and PVP from Sigma-Aldrich, acetonitrile, ammonium formate, formic acid, methanol, and water for LC/MS from J.T. Baker (Phillipsburg, NJ, USA), hydrochloric acid (HCl) and sodium hydroxide (NaOH) for analysis from POCh (Gliwice, Poland), tablets of Metformax^®^ 1000 mg from Teva UK Ltd. (Castleford, UK) and Novonorm^®^ 2 mg from Novo Nordisk (Bagsvaerd, Denmark).

### 3.2. LC-UV Method

#### 3.2.1. Chromatography

Chromatography was carried out while using a 306 model pump and UV170 model detector from Gilson (Middleton, WI, USA). The mobile phase consisted of 0.01 M ammonium acetate and acetonitrile (60:40, *v/v*) and it was pumped with the flow rate of 1.0 mL/min. Separation was achieved on a LiChrospher^®^CN column (125 × 4.0 mm, 5 µm) from Merck (Darmstadt, Germany) that was housed in a column heater set at 25 °C with UV detection at 235 nm. The chromatographic system was controlled by Omnic software from Gilson (Middleton, WI, USA). 

#### 3.2.2. Stock Solutions

The stock solutions were prepared by dissolving metformin and repaglinide in methanol to obtain the concentrations of 1 mg/mL. These solutions were stored for 48 h at 30 °C to confirm their stability, and further analyzed for additional peaks on the chromatograms and for recoveries. All of the working solutions were prepared from these stock solutions by respective diluting with methanol.

#### 3.2.3. Robustness

The robustness of the method was checked by deliberately changing the optimal chromatographic conditions. The flow rate of the mobile phase was changed from 1.0 to 0.8 and 1.2 mL/min. The organic strength of the mobile phase varied by 2% (from 40 to 38 and 42% of acetonitrile) and the UV detection wavelength from 235 to 232 and 238 nm. One factor was changed at the time and, finally, the solutions of metformin and repaglinide (0.05 mg/mL) were applied onto the column in triplicate to estimate the differences in the peak shapes, peak areas, and retention times.

#### 3.2.4. Linearity

The working solutions of metformin and repaglinide were prepared to reach the concentration range from 0.015 to 0.09 mg/mL. Afterwards, six injections were made onto the column for each working solution. The obtained peak areas were plotted against the corresponding concentrations of the drugs to construct the calibration equations. The limit of detection (LOD) and the limit of quantification (LOQ) were determined from the standard deviation of the intercepts and slopes of the calibration lines, while using 3.3 and 10 multipliers for LOD and LOQ, respectively. 

#### 3.2.5. Precision and Accuracy

The precision of the method was evaluated by injecting the solutions of metformin and repaglinide onto the column at concentrations of 0.02, 0.05, and 0.085 mg/mL. The samples were analyzed three times during one day and for three subsequent days. 

The accuracy of the method was estimated by analyzing both drugs in the samples of powdered tablets and comparing the determined amounts to the nominal values. The weighed portions of powdered tablets containing approximately 50 mg of metformin were transferred to 50 mL volumetric flasks with 30 mL of methanol, sonicated for 30 min., diluted to the mark, and then filtered by nylon membrane filters (0.45 μm). The weighed portions of powdered tablets containing approximately 5 mg of repaglinide were transferred to 5 mL volumetric flasks with 3 mL of methanol, sonicated for 30 min., diluted to the mark, and filtered by nylon membrane filters (0.45 μm). Afterwards, 0.5 mL volumes from these solutions were mixed thoroughly, diluted to 10 mL, and then injected onto the column. The assay was repeated six times, individually weighing the respective portions of powdered tablets. Finally, the concentrations of metformin and repaglinide were calculated while using respective calibration equations. The calculated RSD values were used for precision and calculated percentage recoveries for accuracy.

#### 3.2.6. Specificity

The specificity of the method was examined by determining of metformin and repaglinide in the samples that were subjected to degradation under extreme conditions (0.1 M HCl, 0.1 M NaOH and 3% H_2_O_2_ at 70 °C for 240 min.). The specificity of the method was also examined by analyzing metformin and repaglinide in respective tablets, in the presence of the excipients that were used for manufacturing.

### 3.3. Degradation in Solutions

Degradation of metformin and repaglinide was performed according to ICH Q1A guideline [4], in 0.01 and 0.1 M HCl, 0.01, and 0.1 M NaOH, and 0.3 and 3% H_2_O_2_, at 70 °C. The samples of metformin weighing approximately 5 mg were dissolved in 1.0 mL of water and then 2 mL of respective medium was added. Samples of repaglinide weighing approximately 5 mg were dissolved in 1.0 mL of methanol and 2 mL of the appropriate medium was then added. Thus, theoretical concentrations of both drugs were 0.05 mg/mL. The tubes containing the above samples were tightly closed with stoppers and placed in a thermostated water bath from WSL (Warszawa, Poland) set at 70 °C. The respective samples were removed from the bath after every 15 min., while the total time of experiment was 240 min. The samples were cooled, neutralized if necessary, and then diluted to 10 mL with methanol.

#### 3.3.1. Quantitative Analysis after Degradation

Volumes of 1.0 mL from the stressed samples were diluted to 10 mL with methanol and then analyzed by our LC-UV method described above. The procedure was repeated three times for each sample and mean concentrations of the remaining (non-degraded) metformin and repaglinide were calculated from respective calibration equations. At the same time, the percentage levels of the degradation of the drugs were calculated while taking their starting concentrations into account.

The kinetic parameters were calculated when the level of degradation was at least 5% during 240 min. The concentration of non-degraded drugs or logarithm of the concentration of non-degraded drugs were plotted against time of degradation to obtain the equations y = ax+b and the determination coefficients R^2^ and, in consequence, to determine the reaction order. Subsequently, further kinetic parameters, i.e., degradation rate constant (k), the time of degradation of 10% (t_0.1_), and the half life time (t_0.5_) were calculated. 

#### 3.3.2. Parameters of LC-MS Method

LC-MS analysis was performed while using a 6530B Accurate-Mass QTOF spectrometer with a dual ESI-Jet stream ion source, a DAD, and a binary gradient pump system from Agilent Technologies Inc. (Santa Clara, CA, USA).

For metformin, a LiChrospher^®^CN column (125 × 4.0 mm, 5 µm), similarly to our LC-UV method, was used. Water with formic acid (0.2%) (solvent A), and acetonitrile with formic acid (0.2%) (solvent B) were used as the mobile phases with the following gradient programme: 0–35 min., 0–95% solvent B with post time of 5 min. The total time of analysis was 30 min. with a stable flow rate at 0.5 mL/min.

For repaglinide, a Gemini C18 110A column (100 × 2.0 mm, 3 µm) from Phenomenex Inc. (Torrance, CA, USA) were used. Acetonitrile-water (1:99, *v/v*) with 10mM ammonium formate (0.1%) (solvent A) and acetonitrile-water (95:5, *v/v*) with 10mM ammonium formate (0.1%) (solvent B) were used as follows, 0–2 min.: 100% of solvent B; 2–18 min.: 0–90% of solvent B; 18–20 min.: 90% of solvent B; post time 10 min. The total time of analysis was 30 min. with a stable flow rate at 0.4 mL/min. 

ESI-QTOF-MS analysis was performed according to the following parameters of the ion source: dual spray jet stream ESI, positive ion mode, gas (N_2_) flow rate 12 L/min., nebulizer pressure 30 psig, gas temperature 200 °C (metformin) or 300 °C (repaglinide), sheath gas temperature 350°, sheath gas flow 12 L/min., VCap 4000V, skimmer 65 V, fragmentor 150 V, and octopole RF Peak: 750 V, the *m/z* range 50–1000 mass units, with acquisition mode auto MS/MS, the scan rate three spectra per cycle, and collision induced dissociation (CID) of 20 eV. Additional analysis was made in auto MS/MS with excluded *m/z* at 922.0097 and 121.0508, for positive ion mode corresponding to the *m/z* of reference ions. Our apparatus worked in mass error within the range of 0–2 ppm.

### 3.4. Degradation in the Solid State

Binary mixtures containing metformin or repaglinide with four excipients (PVP, mannitol, magnesium stearate, and lactose) were prepared in 1:1 ratio (*w/w*), by weighting respective amounts of the substances and grinding them in an agate mortar. The samples were placed in standardized small flat vessels, such that the layer thickness was approximately 3 mm. These vessels were placed in a climate chamber KBF P240 from Binder (Neckarsulm, Germany) that was set at 70 °C/70% RH, and stored for two months. Similar samples were placed in a Suntest CPS Plus chamber from Atlas (Linsengericht, Germany) and then exposed to UV/VIS light (300–800 nm) with energy equal to 18902 and 56,706 kJ/m^2^. The energy of 18,902 kJ/m^2^ was equivalent to 1,200,000 lux and 200 Wh/m^2^, which is recommended by ICH Q1B guideline to confirm the stability of the drug [45], while the energy of 56,706 kJ/m^2^ was three times higher and it was used for stress degradation. 

### 3.5. FT-IR Method

After finishing the stress experiments in the solid state, the samples were transferred to tightly closed containers and then stored in a dessicator until analysis. The FT-IR spectra were recorded on a Nicolet 6700 spectrometer that was equipped with a Smart iTR accessory from Thermo Scientific (Waltham, MA, USA). After recording a background spectrum, the quantities of approximately 2 mg of the samples were placed on the diamond. Afterwards, four scans were recorded for each sample over the range 4000–800 cm^−1^ with a resolution of 4 cm^−1^. The spectra of stressed metformin and repaglinide, as well as of their mixtures with respective excipients, were compared with those that were obtained for the non-stressed samples. 

## 4. Conclusions

The results that are presented here complement current knowledge regarding the chemical stability of two important antidiabetic drugs, metformin and repaglinide. A few analytical techniques, i.e., LC-UV, LC-MS, and FT-IR, were applied to reach more definite conclusions. For the first time, kinetic parameters were calculated for the degradation processes of metformin and repaglinide in solutions. It was shown that the drugs varied in their stability, i.e., metformin was more sensitive to alkali and less to oxidants, when repaglinide was more sensitive to acids and oxidants. Thus, the necessity of protecting their binary formulations from all of these factors was shown. In our LC-MS study, two new degradation products of metformin and nine new degradation products of repaglinide were identified, and new degradation pathways were proposed. These data may be the starting point for further studies for qualifying them as new related substances in pharmacopoeial monographs. What is more, the chemical stability of both drugs was examined in the presence of excipients with different chemical properties. The necessity of avoiding of PVP, magnesium stearate, and mannitol in manufacturing of formulations containing metformin and repaglinide could be recommended because some of them were shown to be reactive. Thus, the presented results may serve as a starting point for designing new combined formulations of metformin and repaglinide.

## Figures and Tables

**Figure 1 molecules-24-04430-f001:**
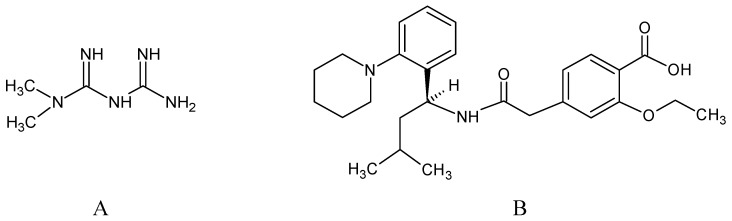
Chemical structures of metformin (**A**) and repaglinide (**B**).

**Figure 2 molecules-24-04430-f002:**
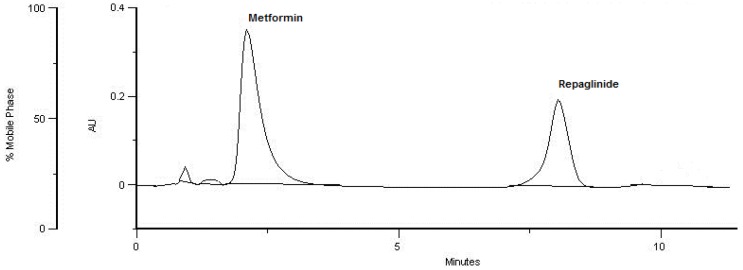
LC-UV chromatogram of metformin (stressed with 0.1 M NaOH) and repaglinide (stressed with 0.1 M HCl).

**Figure 3 molecules-24-04430-f003:**
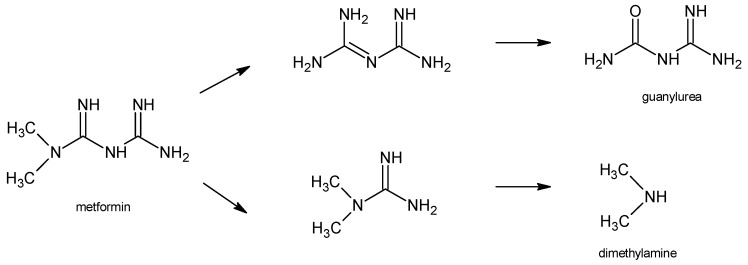
Two proposed degradation pathways of metformin [31].

**Figure 4 molecules-24-04430-f004:**
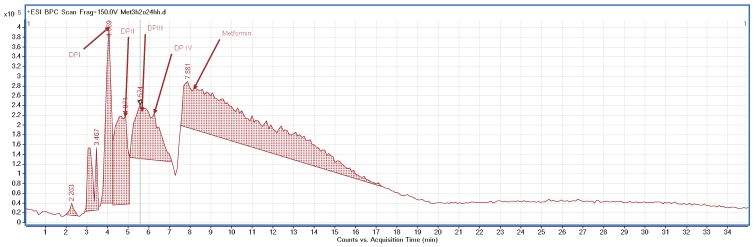
BPC chromatograms of metformin and its degradation products in 3% H_2_O_2._

**Figure 5 molecules-24-04430-f005:**
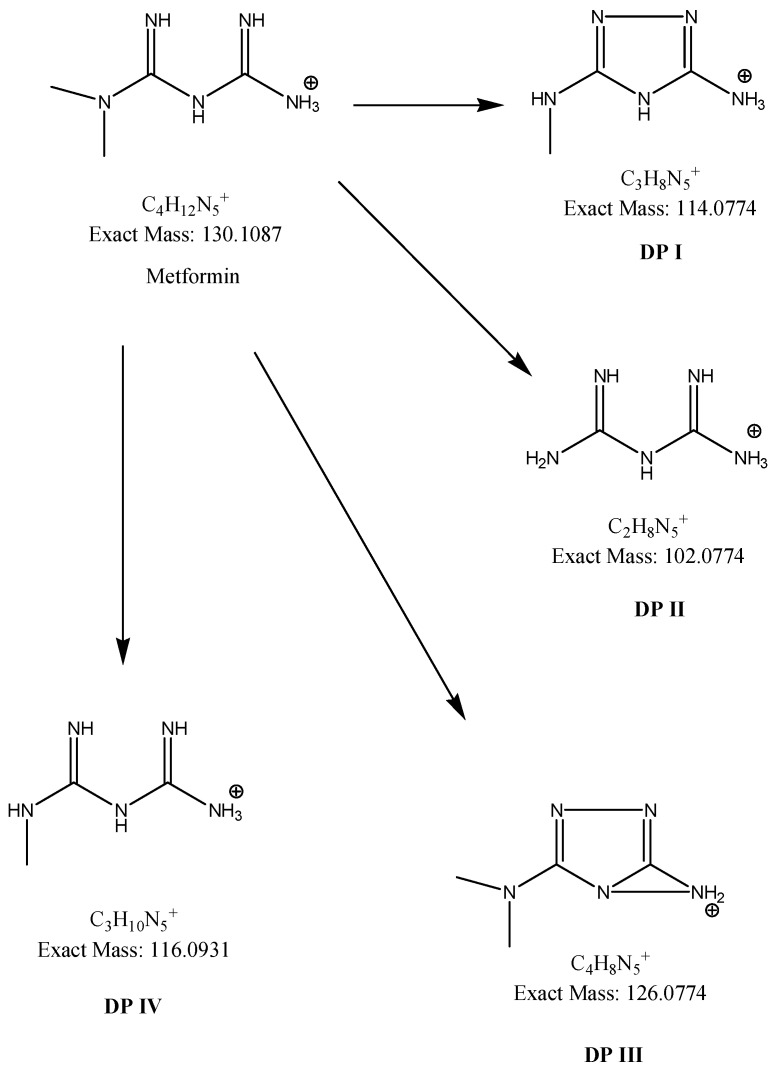
Proposed degradation pathways of metformin in 3% H_2_O_2_ at 70 °C.

**Figure 6 molecules-24-04430-f006:**
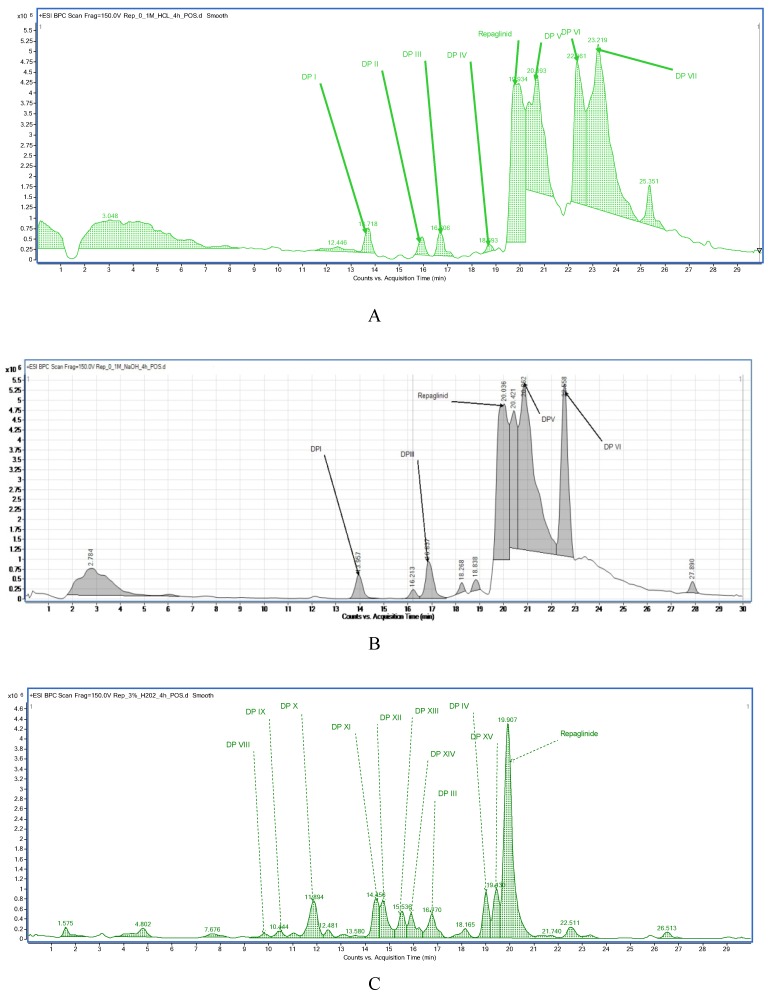
BPC chromatograms of repaglinide and its degradation products: in 0.1 M HCl (**A**), in 0.1 M NaOH (**B**), and in 3% H_2_O_2_ (**C**).

**Figure 7 molecules-24-04430-f007:**
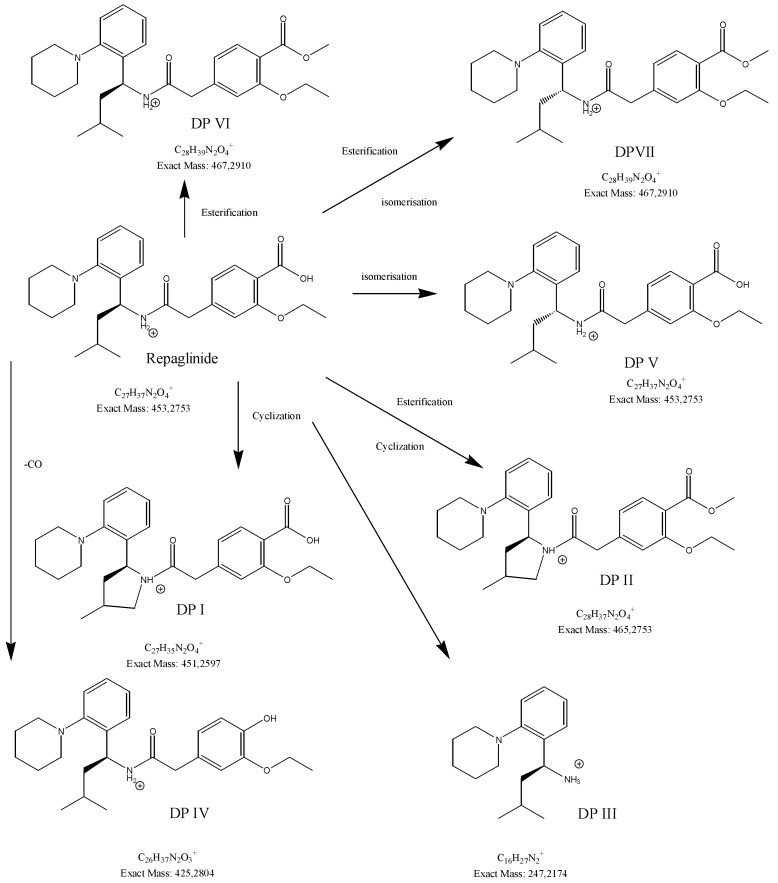
Proposed degradation pathways of repaglinide in 0.1 M HCl at 70 °C.

**Figure 8 molecules-24-04430-f008:**
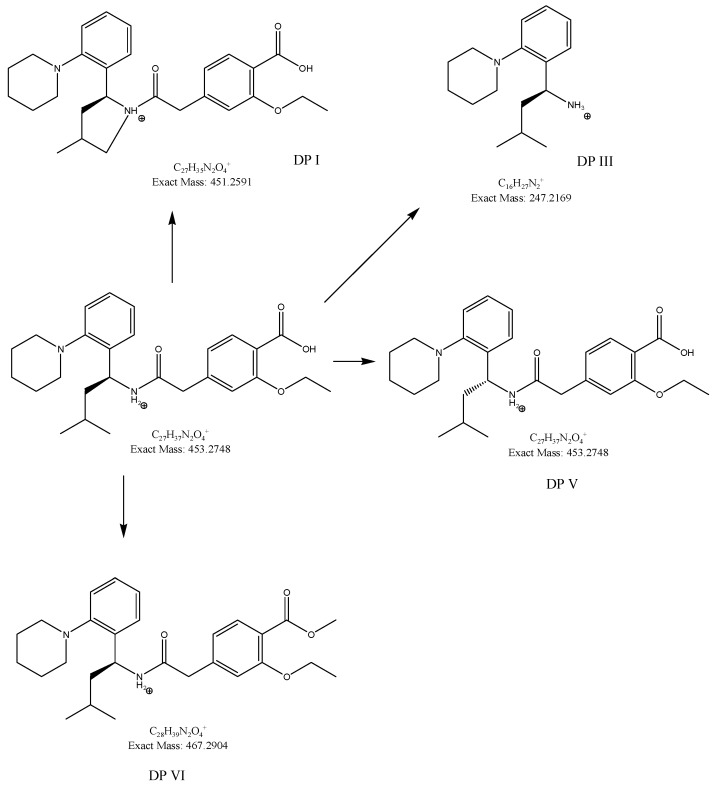
Proposed degradation pathways of repaglinide in 0.1 M NaOH at 70 °C.

**Figure 9 molecules-24-04430-f009:**
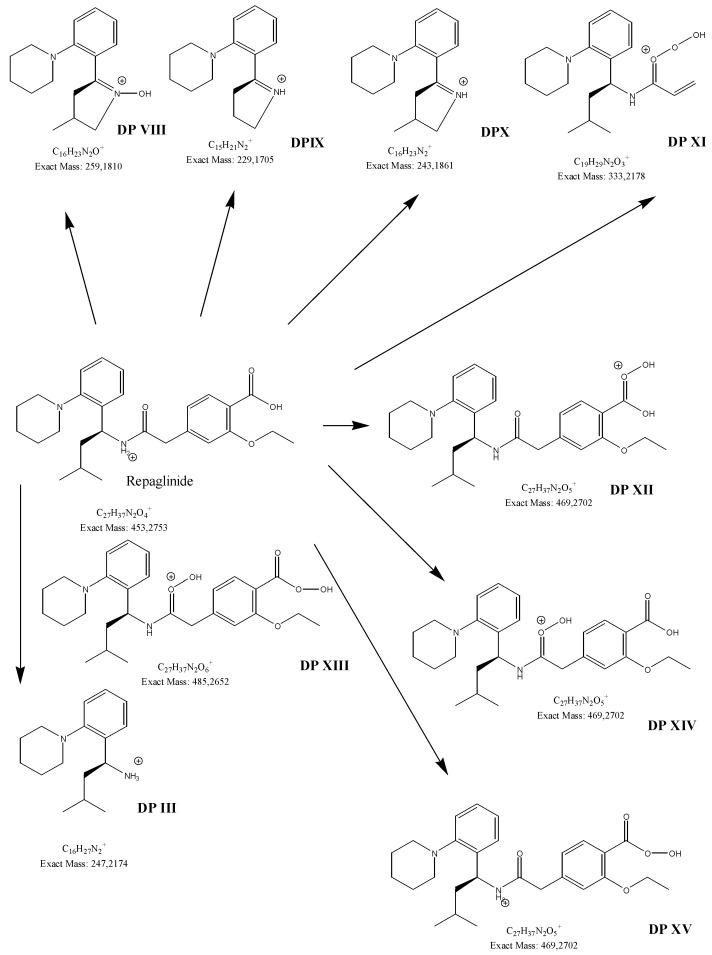
Proposed degradation pathways of repaglinide in 3% H_2_O_2_ at 70 °C.

**Figure 10 molecules-24-04430-f010:**
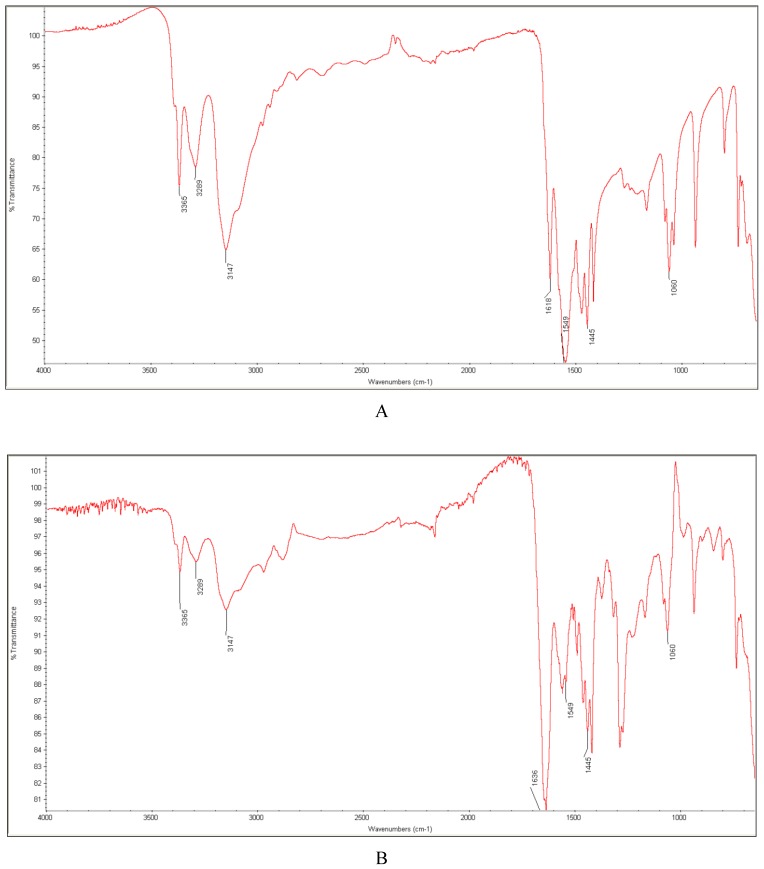
FT-IR spectra of: non-stressed metformin (**A**), the mixture of metformin with PVP stressed with high temperature/humidity (**B**), the mixture of metformin with mannitol stressed with high temperature/humidity (**C**), and the mixture of metformin with magnesium stearate stressed with high temperature/humidity (**D**).

**Figure 11 molecules-24-04430-f011:**
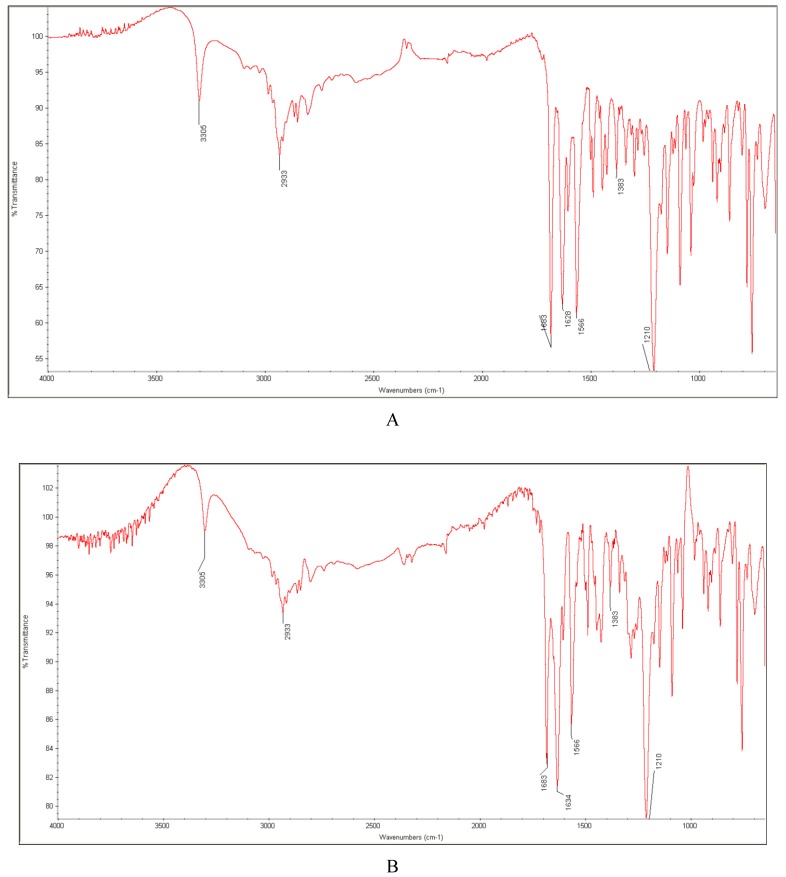
FT-IR spectra of: non-stressed repaglinide (**A**), the non-stressed mixture of repaglinide with polyvinylpyrrolidone (PVP) (**B**), the mixture of repaglinide with PVP stressed with high temperature/humidity (**C**), and the mixture of repaglinide with magnesium stearate stressed with high temperature/humidity (**D**).

**Table 1 molecules-24-04430-t001:** Related substances (Imps A–F) and degradation products (DPs) of metformin.

Compound	[M + H]^+^	Name	Ref.
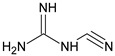 *N*-cyanoguanidine		Imp A	[30]
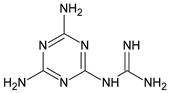 *N*-(4,6-diamino-1,3,5-triazin-2-yl)guanidine		Imp B	[30]
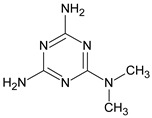 N^2^,N^2^-dimethyl-1,3,5-triazine-2,4,6-triamine		Imp C	[30]
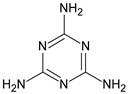 1,3,5-triazine-2,4,6-triamine		Imp D=DP2	[17,30]
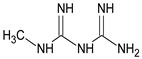 N-methyltriimidodicarbonic diamide (1-methylbiguanide)	116.0931	Imp E=DP3=DPIV	[17,30]
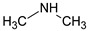 N-methylmethanamine (dimethylamine)		Imp F	[30]
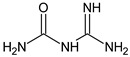 1-carbamimidoylurea (guanylurea)	103	DP1	[31]

**Table 2 molecules-24-04430-t002:** Related substances (Imps A–E) of repaglinide [30].

Compound	[M + H]^+^	Name
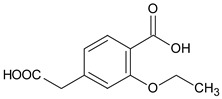 4-(carboxymethyl)-2-ethoxybenzoic acid		Imp A
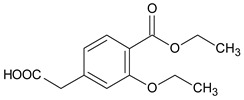 3-ethoxy-4-(ethoxycarbonyl)phenyl]acetic acid		Imp B
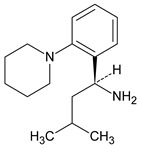 (1*S*)-3-methyl-1-[2-(piperidin-1-yl)phenyl]butan-1-amine	247.2169	Imp C=DPIII

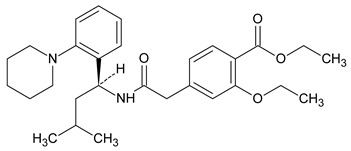 ethyl 2-ethoxy-4-[2-[[(1S)-3-methyl-1-[2-(piperidin-1-yl) phenyl]butyl]amine]-2-ethoxy]benzoate		Imp D
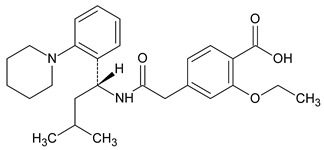 2-ethoxy-4-[2-[[(1*R*)-3-methyl-1-[2-(piperidin-1-yl)phenyl]butyl]amine]-2-ethoxy]benzoic acid (isomer)	453.2753	Imp E=DPV

**Table 3 molecules-24-04430-t003:** Validation of LC-UV method for determination of metformin and repaglinide.

Parameter	Metformin	Repaglinide
t_R_ (min)	2.55	7.68
Asymmetry factor	1.5	0.8
Linearity range (mg/mL)	0.015–0.09	0.015–0.09
Slope	44.4609	9.2725
SD of slope	0.5807	0.0738
Intercept	0.1877	0.0039
SD for intercept	0.0189	0.0017
R^2^	0.9993	0.9995
SD of R^2^	0.0003	0.0003
LOD (mg/mL)	0.001	0.0006
LOQ (mg/mL)	0.004	0.002
Precision (RSD)		
Intra-day	1.07–1.55	0.67–0.95
Inter-day	1.28–1.60	0.82–1.15
Accuracy (%)	99.81–100.98	98.48–101.63

**Table 4 molecules-24-04430-t004:** Robustness of LC-UV method for determination of metformin and repaglinide.

	Metformin t_R_	Metformin Peak Area	Repaglinide t_R_	Repaglinide Peak Area	R_s_
Flow rate (mL/min)					
0.8	2.51	2.43189	7.65	0.46497	7.35
1.0	2.55	2.39286	7.68	0.46730	7.25
1.2	2.53	2.38583	7.65	0.46674	7.21
Acetonitrile (%)					
35	2.53	2.38583	7.70	0.47090	7.23
40	2.55	2.39286	7.68	0.46730	7.25
45	2.56	2.38111	7.62	0.47090	7.25
UV detection (nm)					
232	2.55	2.38320	7.67	0.46672	7.24
235	2.55	2.39286	7.68	0.46730	7.25
238	2.53	2.38229	7.68	0.46719	7.27

**Table 5 molecules-24-04430-t005:** Kinetics of degradation of metformin and repaglinide in solutions.

Conditions	Degradation [%]	y = ax + b	R^2^	K [s^−1^]	t_0.1_ [h]	t_0.5_ [h]
Metformin						
0.01 M HCl	5.73	y = −0.0003x + 1.6154	0.9816	1.15 × 10^−5^	2.55	16.74
0.1 M HCl	6.73	y = −0.0003x + 1.6035	0.9743	1.15 × 10^−5^	2.55	16.74
0.01 M NaOH	9.11	y = −0.0004x + 1.5817	0.9585	1.54 × 10^−5^	1.90	12.50
0.1 M NaOH	60.92	y = −0.0052x + 1.6011	0.9984	1.99 × 10^−4^	0.15	0.97
0.3% H_2_O_2_	6.58	y = −0.0001x + 1.5717	0.9770	3.84 × 10^−5^	7.62	50.13
3% H_2_O_2_	7.95	y = −0.0003x + 1.6076	0.9699	1.15 × 10^−5^	2.55	16.74
Repaglinide						
0.01 M HCl	19.93	y = −0.0012x + 1.6041	0.9748	4.61 × 10^−5^	0.64	4.18
0.1 M HCl	38.32	y = −0.0024x + 1.6435	0.9340	9.21 × 10^−5^	0.32	2.09
0.01 M NaOH	6.13	y = −0.0003x + 1.5762	0.9275	1.15 × 10^−5^	2.55	16.74
0.1 M NaOH	7.24	y = −0.0004x + 1.6013	0.9666	1.54 × 10^−5^	1.90	12.50
0.3% H_2_O_2_	9.16	y = −0.0006x + 1.6033	0.9546	2.30 × 10^−5^	1.09	8.37
3% H_2_O_2_	21.75	y = −0.0014x + 1.6225	0.9691	5.37 × 10^−5^	0.55	3.58

**Table 6 molecules-24-04430-t006:** Degradation products (DPs) of repaglinide [30,37,38].

Compound	[M + H]^+^	Name	Compound	[M + H]^+^	Name
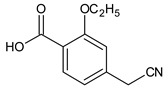	206.2	DP1	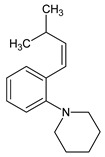	230	DP8
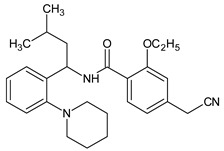	434.4	DP2	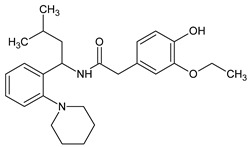	425	DP9
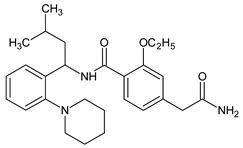	452.5	DP3	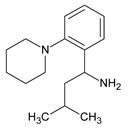	247	DP10=Imp C
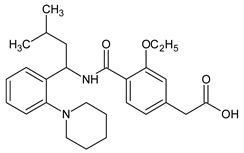	453.5	DP4	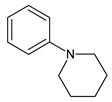	162	DP11
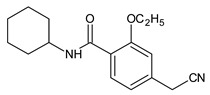	287.4	DP5	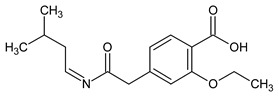	292	DP12
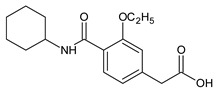	306.4	DP6	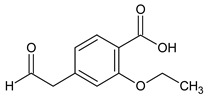	209	DP13
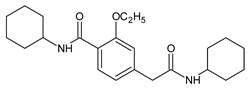	387.5	DP7			

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
