# Peer review of "Determination of Chemical Stability of Two Oral Antidiabetics, Metformin and Repaglinide in the Solid State and Solutions Using LC-UV, LC-MS, and FT-IR Methods"

_molecules, 2019, doi:10.3390/molecules24244430_

Round 1

Reviewer 1 Report

The work is publishable. Since its originality lies in the identification of new intermediates and pathways, the clear identification of the compounds are critical. The author fails to elaborate how the exact masses of intermediate compare with the theoretical exact masses. a list of the comparison and error calculation will make assignment more reliable. In addition the separation of mixture for degradation should be improved.    

Author Response

Thanks for the comments. It is really important to ensure that the ion m/z have been measured with as low mass error as possible in HR mass spec analysis. Our apparatus worked in the mass error within range of 0-2 ppm. All the peaks have been measured with such the error and the respective information was added in our revised text. We did not include the respective data in the manuscript because there are many more important results in it, and we tried to avoid overloading of the data in our manuscript.

Each degradation product was properly separated. What sometimes seems as poor separation baseline is a gradient separation line, but in Extracted Ion Chromatogram mode each peak is well separated from the another one.

English language and style were carefully checked and corrected. Finally, the manuscript was checked by a native English speaking colleague.

Reviewer 2 Report

The authors applied a variety of methods to measure the stability and degradation products of two important oral antidiabeitcs under various conditions, which is an important attribute to the study of these two drugs. Though the authors showed a lot of results, I would recommend the authors spend more efforts on discussion part to make the results and their importance more clear.

Author Response

Thanks for the comment. Our discussion was carefully revised and some new statements on the importance of our results were added, e.g. in the area of potential interactions of metformin or repaglinide and the excipients with different reactivity. As a result, two new references were added in the revised text. In addition, some new sentences were added in the parts concerning new degradation products of metformin and repaglinide. We hope that now our manuscript could be more interesting for the readers.